# Microbiological Surveillance of Endoscopes in a Southern Italian Transplantation Hospital: A Retrospective Study from 2016 to 2019

**DOI:** 10.3390/ijerph18063057

**Published:** 2021-03-16

**Authors:** Valentina Marchese, Daniele Di Carlo, Gaetano Fazio, Santi Mauro Gioè, Angelo Luca, Rossella Alduino, Monica Rizzo, Fabio Tuzzolino, Francesco Monaco, Pier Giulio Conaldi, Bruno Douradinha, Giuseppina Di Martino

**Affiliations:** 1Istituto di Ricovero e Cura a Carattere Scientifico-Istituto Mediterraneo per i Trapianti e Terapie ad Alta Specializzazione (IRCCS-ISMETT), 90127 Palermo, Italy; v.mark@hotmail.it (V.M.); ddicarlo@ismett.edu (D.D.C.); gfazio@ismett.edu (G.F.); smgioe@ismett.edu (S.M.G.); aluca@ismett.edu (A.L.); ralduino@ismett.edu (R.A.); morizzo@ismett.edu (M.R.); ftuzzolino@ismett.edu (F.T.); fmonaco@ismett.edu (F.M.); pgconaldi@ismett.edu (P.G.C.); gdimartino@ismett.edu (G.D.M.); 2Fondazione Ri.MED, 90127 Palermo, Italy

**Keywords:** endoscopes, reprocessing, microbiological surveillance, gram positive, gram negative, transplantation

## Abstract

Endoscopes are medical instruments that are used routinely in health structures. Due to their invasive nature and contact with many patients, they may cause hospital-acquired infections if not disinfected correctly. To ensure a high-level disinfection procedure or reprocessing, since the methods currently adopted in our institute are adequate, we evaluated retrospectively the presence of microorganisms in our endoscopes after reprocessing. Microbiological surveillance was performed from January 2016 to December 2019 in the instruments in use in our endoscopic room after reprocessing. In total, 35 endoscopes (3 duodenoscopes, 3 echoendoscopes, 12 bronchoscopes, 5 colonoscopes, and 12 gastroscopes) were evaluated for the presence of microorganisms, including multidrug-resistant pathogens and indicator microorganisms (IMOs). Our procedures were in agreement with an internal protocol based on Italian, international, and the Center for Disease Control and Prevention (CDC) recommendations. Of a total of 811 samples, 799 (98.5%) complied with the regulatory guidelines, while 9 (1.1%) were positive for IMOs, and 3 (0.4%) displayed more than 10 colony-forming units (CFU) of environmental and commensal pathogens. Our results show that the internal reprocessing protocol is very efficient, leading to a very low number of observed contaminations, and it could be easily implemented by other health facilities that face a huge number of hospital-acquired infections due to incorrectly disinfected endoscopes.

## 1. Introduction

Endoscopic procedures subject patients to several risks, including the spreading of infectious pathogens. Endoscopes are classified as semi-critical devices, coming into contact with mucosal areas or non-intact skin [1]. In order to reduce the risk of cross-contamination between patients, endoscopes are submitted to a high-level disinfection procedure called reprocessing. The reprocessing of endoscopes includes cleaning of all external and internal surfaces, including the elevator recess, followed by a microbicidal treatment, to ensure that the instrument will be free of any potential pathogenic transmission. Several national and international scientific organizations have defined guidelines regarding the reprocessing of endoscopes [2,3,4,5]. 

To date, the risk of cross-infection with contaminated endoscopes is very low (1 for 1,800,000 endoscopic procedures), as confirmed elsewhere [6,7,8]. However, given the high number of endoscopic procedures performed worldwide on a daily basis, the most frequent medical device-related infections are associated to the use of contaminated endoscopes that underwent a deficient process of cleaning and disinfection prior to use [9,10]. However, available data concerning the levels of contamination of endoscopes are often not very comparable among them, due to the lack of standardized sampling procedures in the guidelines and well-defined sampling sites, criteria, and contamination cut-offs. Important cross-transmission outbreaks of multidrug-resistant bacteria have been reported following the use of contaminated duodenoscopes due to the failure of automatic reprocessors or insufficient application of reprocessing procedures. These outbreaks resulted in serious infections of carbapenem-resistant *Enterobacteriaceae*, *Escherichia coli*, and *Klebsiella pneumoniae*, such as septicemia, biliary disease, and urinary infections, and mortality rates of up to 56% of patients who had been submitted to at least one endoscope procedure [11,12,13,14,15,16,17]. Infections after endoscopic retrograde cholangiopancreatography procedures typically vary between 2 and 4% [9]. In addition, a recent report from an Italian teaching hospital showed that 37.8% of the endoscopes, after reprocessing, did not meet the compliance standards indicated by both national and international organizations [18]. Moreover, some studies have demonstrated the persistent presence of microorganisms in endoscopes after reprocessing. Ofstead and colleagues have observed the presence of bacterial pathogens in endoscopes ready to be used on the patient, up to 64% after high-level disinfection (HLD) and up to 9% after storage, in an American hospital [19]. There is evidence that the success of efficient reprocessing of endoscopes depends on correct staff training and of an active, effective microbiological surveillance protocol, which may reduce the cross-transmission of potential contaminations [5]. As an example, two different reports indicate skin bacteria contaminations of 11% [20] and 35% [21] respectively, after reprocessing, in which the authors attribute the contaminations to deficient sampling by the user and not to the disinfection process.

In this study, we evaluated the effectiveness of reprocessing in our health facility, a transplantation Southern Italian hospital, the Istituto Mediterraneo per Trapianti e Terapie ad Alta Specializzazione (IRCCS-ISMETT), through a retrospective analysis of the results of the microbiological surveillance in the period between January 2016 and December 2019. We also considered the advantages and disadvantages related to the implementation of this type of monitoring. Reprocessing effectiveness was assessed by evaluating the contamination rate of the endoscopes, either globally or by type of instrument. 

## 2. Materials and Methods 

### 2.1. Endoscopes

The study was conducted by analyzing the laboratory results of microbiological tests performed on 35 endoscopes (3 duodenoscopes, 3 echoendoscopes, 12 bronchoscopes, 5 colonoscopes, and 12 gastroscopes; Olympus Italia Società a responsabilità limitata (S.r.l.), Segrate, Italy) in the aforementioned 48 months of surveillance. Only instruments used in the endoscopic room were included in the study.

### 2.2. Reprocessing

At IRCCS-ISMETT, approximately 4000 endoscopic procedures are performed every year. The endoscopes are reprocessed by especially trained dedicated personnel, following an in-house validated procedure in line with national [22,23], international [2,3,4,24,25], and with the latest recommendations of the Center for Disease Control and Prevention (CDC) [5]. The endoscopes are disinfected in Steelco EW2 reprocessors (Steelco S.p.A., Riese Pio X, Italy) within our endoscopy service. They are pass-through washers installed between the “dirty” and “clean” areas of the room. The disinfection process is based on the use of peracetic acid. For each cycle, the endoscope washing machine issues a traceability ticket. The disinfected endoscopes are placed in special, dedicated storage cabinets where they are subjected to a drying phase and stored, being ready for further use within the following 5 days. Once this time limit has been exceeded, the devices must be reprocessed before further use. The scheme of the reprocessing procedure can be observed in Figure A1 (Appendix A).

### 2.3. Microbiological Sampling

Periodic microbiological sampling is done in all endoscopic instruments in use in our endoscopy service, operating rooms, and in intensive care. Samplings are done as indicated in CDC guidelines for quantitative analysis [5] and in the Queensland operating protocol [26], which advise monthly sampling frequencies for bronchoscopes and duodenoscopes and every three months for all other gastrointestinal endoscopes. Microbiological sampling is done by injecting 50 mL of sterile Dulbecco’s phosphate buffer saline (PBS)/modified without calcium and without magnesium (GE Healthcare, Logan, UT, USA) inside the endoscope duct proximal extremity and collected inside a sterile 50 mL tube and taken to the Diagnostics Unit to be analyzed. It will be centrifuged at 4000× *g* for 10 min, and the pellet will then be resuspended and plated in 2 Columbia Agar with 5% Sheep Blood plates (BD, Heidelberg, Germany). The plates are incubated at 35 °C and 5% CO_2_ and examined every 24 h for 3 days. If after 3 days, no colonies are observed, no growth will be reported. Otherwise, the number of colonies present on the plates will be quantified and identified, and antibiotic susceptibility tests will be run, as previously described [27,28,29]. We assessed the plates for indicator microorganisms (IMOs), which, for reprocessing, are Gram-negative bacteria such as *Escherichia coli*, *Klebsiella pneumoniae*, or other *Enterobacteriaceae* such as *Pseudomonas aeruginosa*, Gram-positive bacteria, such as *Staphylococcus aureus*, beta-hemolytic *Streptococci*, and *Enterococcus* species and fungi and molds, such as *Candida* species or *Aspergillus niger*. A sample is defined as positive if the number of colony-forming units (CFU) is equal to or greater than 10 contaminants (environmental microorganisms or deriving from microbiota of the patient) or if even a single colony of an IMO was observed [2,3,4,5,22,23,24,25].

### 2.4. Report and Follow-Up

The results of the microbiological sampling are sent monthly to the Infection Diseases and Endoscopy departments. They will also be promptly informed in the case of detection of positive samples, specifying which endoscope is contaminated. In the latter case, it is necessary to repeat the reprocessing of the instrument, resample it, and put it in quarantine for 3 days, waiting for the evaluation of the microbiological samples (Figure 1).

In the case of positivity of the instrument due to the presence of IMOs, an epidemiological investigation will be done to identify the patients on whom the endoscope was used and to check for the possible cross-transmission of microorganisms. The instrument can be reused only if the second microbiological sampling is negative. If positivity persists, it is necessary to verify that the recommendations indicated by the manufacturer regarding reprocessing were meticulously followed. 

### 2.5. Statistical Analysis

These data were recorded on an electronic spreadsheet (Excel; Microsoft Redmond, WA, USA). Statistical analysis was carried out using R version 4.0.2 (The R Society, Auckland, New Zealand), SAS 9.4 (Cary, NC, USA), and Microsoft Excel 2013 (Microsoft, Redmond, WA, USA). The outcome variable is defined as a dichotomous variable that describes the presence of positive endoscopes throughout time. Fisher’s exact test was used to compare the presence of contamination between the various types of devices used. The parametric χ^2^ and median tests and the Wilcoxon non-parametric test were used to test the association between endoscope contamination and the number of procedures performed at the time of sampling. Explorative analyses were also done using the Cochran–Armitage Trend Test. Given the presence of repeated measurements, Generalized Estimating Equations (GEE) models were applied to evaluate the effect of time on the outcome. In particular, for the estimation of the model parameters, the correlations between the repeated measures of the outcome variable were considered. Two types of analyses were performed, in which the first considers the contaminations over the years, without however classifying the endoscopes by type, and using an exchangeable correlation working matrix, and a second approach, which considered the contaminations over the years for each observed endoscope, using an autoregressive correlation working matrix. The matrices approaches were chosen based on the quasi-likelihood under the independence model criterion (QIC) obtained, which allows the selection of a proper working correlation structure. Levels of significance were set with *p* < 0.05.

## 3. Results

During the study period (January 2016 to December 2019), 811 samplings were done in a total of 35 endoscopes. All endoscopes were sampled immediately after reprocessing and before proceeding to the next phase, i.e., storage. The samplings showed that nine out of 811 were contaminated with IMOs (1.1%), while three had levels of environmental or commensal contaminants equal or superior to 10 CFU (0.4%), as observed in Table 1. 

On the other hand, 35 samplings showed levels of contaminants inferior to 10 CFU (4.3%), and 764 displayed no contamination at all (94.2%), which means a total of 98.5% samplings were compliant with regulatory guidelines (Table 1). The positivity of a sampling was defined as possessing levels of contamination that were either equal to or superior than one CFU of IMOs or equal to or superior than 10 CFU of contaminants. For all positive samples, it was necessary to repeat the reprocessing, proceed to a second sampling, and put the instruments in quarantine, waiting for the result. The second samplings performed always provided negative results.

Of all tested endoscopes, the higher levels of contaminations were observed in the gastroscopes and in the bronchoscopes (Table 2 and Table 3). 

Although a high level of positivity was observed for duodenoscopes in 2017 (Table 2 and Table 3), no positive samplings were detected in the remaining years in which the study occurred. Regarding the 811 samples taken, gastroscopes are the instruments in which bacterial growth occurs most frequently (4.9%), followed by bronchoscopes (1.4%) (Table 2). However, this statistically significant result (*p* = 0.02) considers the overall presence of microorganisms without discriminating between pathogens, contaminants, and number of CFUs. By analyzing the type of contamination present, in fact, this significance is lost, both considering the presence of IMOs (*p* = 0.11) and evaluating the presence of contaminants with CFU > 10 (*p* = 0.34). Finally, there are no statistically significant results regarding the possible association between samples with microbial growth and the number of procedures performed at the time of sampling (*p* = 0.57). This was confirmed with a further analysis of the 12 cases of positive endoscopes (gastroscopes, bronchoscopes, and duodenoscopes). Using the Cochran–Armitage Trend Test, its outcome showed that the year that recorded the most cases was 2017 and that the association of the latter with the outcome for the three types of endoscopes that showed positivity was not statistically significant. To evaluate how time affected the outcome, a first GEE approach was done using an exchangeable correlation working matrix, in which the comparison between the 4 years showed a statistically significant difference compared to the outcome between 2017 and 2019 (*p* = 0.0292) and between 2016 and 2018 (*p* = 0.0215). The probability of having contaminations in 2016 is 1.27 times higher than in 2018. Similarly, the probability of finding contaminants in 2017 is 4.64 times higher than in 2019. However, since the type of endoscopes was not considered, we did a second GEE analysis, considering a model for each observed endoscope. However, the only instruments with repeated measurements for which it was possible to apply this model with respect to a single endoscope were the gastroscopes, which have shown positive instruments in all the years of the considered study period (Table 3). For the other endoscopes, it was not possible to estimate a model that considered repeated measurements due to convergence problems. Using a GEE model with an autoregressive correlation matrix of order 1 (autoregressive correlation working matrix), we observed that comparison between the 4 years showed a statistically significant difference compared to the outcome between 2016 and 2018 (*p* = 0.033). In particular, the probability of observing contaminations in 2016 was 1.56 times higher than in 2018.

Among the detected microorganisms in the endoscopes (specified on Table 4), IMOs levels contributed greatly to the overall positivity levels per year (Figure 2). 

Out of 12 positive samples, nine were due to IMOs and three were due to environmental or commensal microorganisms (Table 1). In one bronchoscope, in 2019, both IMOs and contaminants CFU equal to 10 were detected (data not shown). For all purposes, this instrument was considered positive due to the presence of IMOs.

## 4. Discussion

Our institute performs around 4000 endoscopic procedures annually, which adds to the need of constant and efficient microbiological samplings of the endoscopes. The Endoscopy service has been equipped with automated endoscope washers Steelco EW2, which can trace all procedures through a printed ticket, but also to keep in memory the entire HLD cycle, thus identifying details regarding the endoscope, date, start and end time of cycle, the outcome of the same, and to record any potential malfunctions.

Several authors have shown a significant reduction in microbial contamination with only manual washing [30], and the recommendations require that cleaning be carried out immediately after use of the endoscope [31,32,33]. Therefore, steps were taken to verify that the recommendations regarding manual washing and those indicated by the manufacturer regarding reprocessing had been scrupulously applied by the personnel in charge. 

During the study period, the culture results showed 12 positive samples, of which nine showed growth of IMOs (Table 1). Therefore, they have all been reprocessed and resampled, being negative after the second sampling. In parallel, the epidemiological investigations of the case were initiated, as per the institute procedure. In the samples showing positivity for pathogenic or commensal germs of the bacterial flora resident with CFU equal to or higher than 10, it was not possible to identify precisely the reason for the lack of effectiveness of the disinfection. The presence of these environmental microorganisms could lead to contamination during the collection of the sample, rather than being due to an error during reprocessing, as stated by the Gastroenterological Society of Australia (GESA) [3,34]. However, the presence of environmental or commensal contaminants but with CFU lower than 10 CFU was considered an absence of growth, as established by the CDC [5], and therefore, no corrective measures or further laboratory investigations were followed. Moreover, epidemiological investigations carried out when the samples tested positive for IMOs excluded in all observed cases a possible cross-transmission between patients through the contaminated instruments (data not shown).

In our study, the final results were obtained 72 h after the incubation of the samples, which was a time when 34.3% (12 out of 35) of the instruments were contaminated. Some studies confirm that this incubation time is necessary to avoid false negatives and report that 30–45% of endoscopic samples become positive after 2 days of incubation [19,35]. In addition, the use of blood agar plates and incubation at 35 °C, as implemented in our protocol, improve the microbiological yield and allow the growth of different microorganisms [19,35]. Analyzing the positivity results in relation to the type of endoscope used, a greater association of the same with gastroscopes is shown, which is difficult to compare with others present in the literature, since few studies have been published regarding this topic [36,37,38]. Our studies showed a positivity rate for gastroscopes of 4.9% (Table 2), if considering all tested endoscopes, or of 35.5%, if considering only the 12 gastroscopes present in-house (Table 3). Both positivity rates are higher than what was reported previously for gastroscopes (1.9%) [38]; however, the authors performed samplings in a much higher number of gastroscopes (1376) than us, which may account for their low positivity rate. Interestingly, the same authors reported a positivity rate of 1.8% for colonoscopes [38] which, in our case, were always negative throughout the study period (Table 2 and Table 3). Similarly to the gastroscopes, we observed a positivity rate for bronchoscopes of 1.4% (Table 2; when all 811 samplings are considered) and of 41.7% if considering the number of bronchoscopes tested, i.e., 12 (Table 3). A recent study in another Italian hospital indicated a positivity rate of 9.8% of their tested 41 bronchoscopes [18]. Due to the difference of the number of the tested bronchoscopes and, since they only considered IMOs as positivity indicators, no direct relationship can be established between their and our results. A recent review on duodenoscopes HDL showed rates of positivity ranging from 1.2% up to 40%, including the presence of multidrug-resistant pathogenic microorganisms [39]. In our study, we observed a positivity rate overall of 0.9% (Table 2) and, if considering only the three duodenoscopes present in our facilities, a positivity rate of 33.3%; i.e., IMOs were detected only in one instrument out of three (Table 3). Thus, our results are in agreement with what has been observed in other health facilities regarding duodenoscopes’ reprocessing. Concerning linear echoendoscopes, one study reported a positivity rate of 57.1% in North American facilities [10], while a Dutch national surveillance [40] reported a positivity rate of 14%. On the other hand, during the study period, we only observed contaminations with environmental microorganisms (with CFU inferior to 10) in the echoendoscopes present at our institute (Table 1), thus being considered compliant with the adopted guidelines [2,3,4,5].

As mentioned above, our reprocessing and microbiological sampling protocol is based on both national and international regulations [2,3,4,5]. Since there are several guidelines, each hospital can adapt its internal procedure to those more relevant to its epidemiological scenario and current needs. For example, European regulations state that a presence of environmental contaminants up to 20 CFU (endoscope channel sample) or up to 100 CFU (washing water) are acceptable [2], and some Northern Italy hospitals have implemented such procedures [41,42]. However, at IRCSS-ISMETT, where many organ transplants are performed each year and the patients are subjected to immunosuppressive regimens, we decided to follow the CDC suggestion [5] and considered only acceptable the presence of commensal microorganisms up to 10 CFU (either from an endoscope channel or from washing water), to avoid potential systemic infections that could be deleterious to the patients. We believe that their protocol also leads to efficient endoscopes’ disinfection but, due to the particular nature of our patients, we prefer to follow stricter guidelines. The choice of suitable reagents used before and during the reprocessing is important to guarantee the maximum disinfection of the endoscopes. Cottarelli et al. performed a microbiological surveillance to assess the level of contamination of their endoscopes after reprocessing [18]. Although their strategy was similar to ours, they lacked uniform reprocessing protocol; i.e., each Endoscopy unit established its own protocol, used different types of reprocessing (manual or automatic), used different reagents for washing and reprocessing, etc. We believe the lack of a standard reprocessing procedure led to the high levels of IMOs found in their endoscopes. On the other hand, we used always the same reprocessing protocol and reagents (Figure A1, Appendix A) that allowed us to achieve very low levels of IMOs and other microbial contaminants, confirming the efficacy of our approach. In addition, we observed a significant decrease in contaminations; i.e., the probability of finding contaminants in the later years (2018 and 2019) was lower than in the first 2 years. We believe that this is probably due to the fact that when implementing a new routine, both minor optimizations and the time to fully adjust to the new protocols are always required. The significant decrease in contaminants load in the later years confirmed that the adoption of our reprocessing protocol was correctly done and that it is efficient.

Some authors showed that the risk of contamination is particularly reduced when endoscopes are placed in storage cabinets and the sampling is performed after this step [35,43]. At the moment, we cannot confirm this, since the sampling of the endoscopes, in our study, was carried out immediately after reprocessing. It would also be interesting to sample multiple sites of the instrument or use different methods, e.g., adenosine triphosphate (ATP) measurement, since different studies have shown that those lead to a different degree of contamination in the tested instrument [44,45].

As shown before by others, the age of the instruments and number of procedures performed does not seem to affect the positivity rate of the endoscopes [36]; thus, we did not take those factors into consideration while performing our analysis.

Our study has demonstrated the importance of microbiological surveillance in the control of nosocomial infections by monitoring the entire disinfection process of endoscopes. Since the preliminary phase of the automated reprocessing is of paramount importance to the elimination of potential microbiological contaminations, we intend to introduce a further step that involves measuring the organic load in the form of ATP. This will serve to verify whether manual washing has been effective, confirming that the reduction of the organic material would be below a precise cut-off required to proceed further, i.e., to automatically reprocess the endoscopes. This methodology has already been implemented in other facilities and its effectiveness has been evaluated positively [45,46,47].

## 5. Conclusions

Our study showed that the sampling, reprocessing, and surveillance methods adopted by our institute are very efficient and could be easily applied to other health facilities to avoid the potential spreading of multidrug-resistant pathogens. In addition, they are in agreement with national and international guidelines, meaning that they can be used by international institutes, which perform routinely endoscopic procedures. We believe we can still improve our protocol using novel bioassays, e.g., measuring ATP to assess organic matter in the endoscopes, after manual washing. Another limitation of our study is the range of pathogens that can be detected. To detect contamination, our protocol foresees an incubation at 35 °C for 3 days. However, some yeasts and mold may need 4 or 5 days to grow, while other microorganisms have different optimal growth temperatures. Thus, such microorganisms, in our current setup, would not be detected. This strengthens the importance of knowing the epidemiological scenario that characterizes the health facility where this type of protocol is needed and if it requires optimization to identify particular pathogens. In close collaboration with our Diagnostics Unit, we are performing an epidemiological surveillance to understand the nature and antibiotic resistance of pathogenic microorganisms that circulate in our institute. This information will help us adapt our protocols for the detection of novel contaminants that may be present in endoscopes and stop their further spreading to other patients. 

## Figures and Tables

**Figure 1 ijerph-18-03057-f001:**
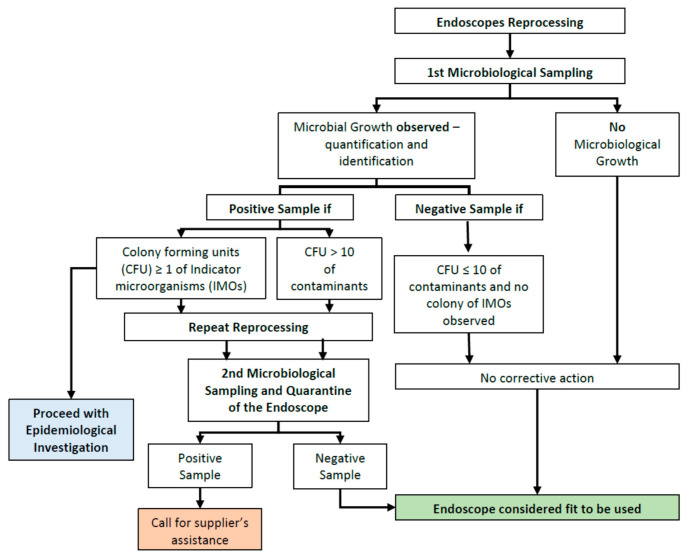
DScheme of actions and procedures to be followed after microbiological sampling of reprocessed endoscopes currently adopted at Istituto Mediterraneo per Trapianti e Terapie ad Alta Specializzazione (IRCCS-ISMETT).

**Figure 2 ijerph-18-03057-f002:**
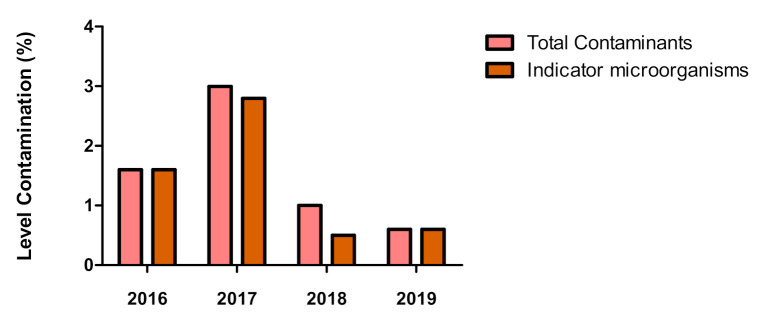
Levels of total contaminants and Indicator microorganism (IMOs) per year.

**Table 1 ijerph-18-03057-t001:** Number of total samplings by instrument and number of indicator microorganisms (IMOs) and contaminants, with respective level of contamination in percent, from January 2016 to December 2019.

Type of Endoscope	Instruments (*n*)	Samplings (*n*)	Indicator Microorganisms (IMOs) ≥ 1 (%)	Contaminants ≥ 10 (%)	Contaminants < 10 (%)	No Growth (%)
Duodenoscopes	3	114	1 (0.9%)	0 (0.0%)	3 (2.6%)	110 (96.5%)
Echoendoscopes	3	117	0 (0.0%)	0 (0.0%)	2 (1.7%)	115 (98.3%)
Bronchoscopes	12	370	3 (1.0%)	1 (0.3%)	22 (5.9%)	344 (93.0%)
Colonoscopes	5	67	0 (0.0%)	0 (0.0%)	1 (1.5%)	66 (98.5%)
Gastroscopes	12	143	5 (3.5%)	2 (1.4%)	7 (4.9%)	129 (90.2%)
Total	35	811	9 (1.1%)	3 (0.4%)	35 (4.3%)	764 (94.2%)

**Table 2 ijerph-18-03057-t002:** Number of microbiological samplings and level of contamination (%) of the disinfected endoscopes.

Type of Endoscope	2016	2017	2018	2019	Total
Duodenoscopes	29 (0.0%)	17 (5.9%)	26 (0.0%)	42 (0.0%)	114 (0.9%)
Echoendoscopes	28 (0.0%)	26 (0.0%)	29 (0.0%)	34 (0.0%)	117 (0.0%)
Bronchoscopes	72 (1.4%)	78 (3.8%)	102 (0.0%)	118 (0.8%)	370 (1.4%) ^†^
Colonoscopes	14 (0.0%)	13 (0.0%)	18 (0.0%)	22 (0.0%)	67 (0.0%)
Gastroscopes	29 (6.9%)	35 (5.7%)	41 (4.9%)	38 (2.6%)	143 (4.9%) ^†^
Total	172 (1.6%)	169 (3.0%)	216 (1.0%)	254 (0.6%)	811 (1.5%)

^†^*p* = 0.02, considering all contaminants, *p* = 0.11 considering only IMOs, *p* = 0.34 considering only contaminants with colony-forming unit (CFU) > 10.

**Table 3 ijerph-18-03057-t003:** Number of positive endoscopes between January 2016 and December 2019.

Type of Endoscope	2016	2017	2018	2019	Total
Duodenoscopes	0	1	0	0	1
Echoendoscopes	0	0	0	0	0
Bronchoscopes	0	3	0	1	5
Colonoscopes	0	0	0	0	0
Gastroscopes	2	2	2	1	7
Total	2	6	2	2	12

**Table 4 ijerph-18-03057-t004:** IMOs and contaminants in colony forming units (CFU ≥ 10) found in the endoscopes tested at Istituto di Ricovero e Cura a Carattere Scientifico – Istituto Mediterraneo per i Trapianti e Terapie ad Alta Specializzazione (IRCCS-ISMETT) from January 2016 to December 2019.

Indicator Microorganisms (IMOs)	Contaminants
*Pseudomonas aeruginosa* *Klebsiella pneumoniae* *Candida parapsilosis* *Candida glabrata* *Aspergillus niger* *Enterococcus faecalis* *Candida tropicalis* *Enterobacter cloacae* *Pseudomonas putida*	Coagulase-negative *Staphylococci* *Moraxella osloensis* *Rothia mucillaginosa*

## Data Availability

Data is contained within this article.

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
