# Peer review of "Microbiological Surveillance of Endoscopes in a Southern Italian Transplantation Hospital: A Retrospective Study from 2016 to 2019"

_ijerph, 2021, doi:10.3390/ijerph18063057_

Round 1
Reviewer 1 Report
The manuscript has been significantly improved and now warrants publication in IJERPH
Reviewer 2 Report
The manuscript reads in a very high quality form after the resubmission.
This manuscript is a resubmission of an earlier submission. The following is a list of the peer review reports and author responses from that submission.
Round 1
Reviewer 1 Report
Manuscript ID: ijerph-1025810-peer-review-v1
Title: Microbiological surveillance of endoscopes in a Southern Italian transplantation hospital: a retrospective study from 2016 to 2019
Authors: Bruno Douradinha * , Valentina Marchese , Daniele Di Carlo , Gaetano Fazio , Santi Mauro Gioè , Angelo Luca , Rossella Alduino , Francesco Monaco , Pier Giulio Conaldi , Giuseppina Di Martino
General Comments:
Authors provide a unique research article discussing an important topic of surveillance of medical instrument infection. This research is really important, especially given the rise of antimicrobial resistant bacteria within community, clinical, and environmental settings. It is also an interesting question of evaluating different medical devices as potential reservoirs for bacterial fomites. It appears that the almost all devices exhibit no growth with the exception of gastroscopes and bronchoscopes. In regard to these two devices, I would strongly encourage the authors to entertain the idea of using more advanced statistical analyses to disentangle contamination burdens over time with. Additionally, the introduction can be shortened and expand upon statistical methods. The results are somewhat challenging to follow given the enormity of the number of tables. The figures and tables would be greatly improved upon by using grey scale and color when appropriate.
Detailed Comments:
- Clarify impacts on mortality and morbidity instead of “poor outcome” in lines 53-54.
- Provide more detail in “persistent contamination” in line 59 as contamination can occur as both a chemical, microorganism, etc.
- Figure 1 has too much text. I would suggest either making a flowchart with visuals or moving this to the appendices. Also, from an aesthetic standpoint be mindful of color use; the arrows can likely just be black.
- Section 2.3 microbiological sampling – It looks like most of this is repeat from Figure 1. I would shorten this section to make it more readable and only highlight important methodologies that are relevant to the experimental design.
- Report and follow up – The quarantine time should be specified. Again, this text also needs to be reduced (e.g. you don’t need to state “"medical head and head nurse of the endoscopy service and head nurse of the infection control service. 132" – simply include the official title of the clinical lab department that processes samples).
- It would also strengthen the paper if the authors could clearly outline the study aim and predictions before reaching the methods. Based off the methods, the results refer more to point estimates than “assessed by estimating the contamination rate”. Line 77-80 starting with “Our studies showed…” needs to be omitted. This is language used in a results section not the introduction.
- Lines 189-190: there is a lot of experimental design and statistical analysis assumptions of basing the results off one Fisher statistic. These types of exploratory analyses should only be used for descriptive statistics but not to discern any effect or association. It would greatly benefit the manuscript to clearly define your outcome variable and predictor (y, time period?). It looks like your outcome variable is a binary categorization of contamination presence or absence based on the previously defined instrument review criteria.
- From an epidemiological perspective, paragraph 141-148 is the most important of the paper. Your definitions of IMOs and other contaminates need to be clearly defined. For example, if you
- Lines 191-193: These points made here are concerning of changing the experimental questions to stratify the sampling and minimize bacterial contamination and increase the p-value. Normally, it is not appropriate to have discussion of the results in this section, but it’s also hazardous to make assumptions about the level of risk these very aggregated definitions of bacterial groups.
- Lines 193-195: The question of repeated samples really needs to evaluated with statistical models and preferably hierarchical models to control for clustering over time.
- Line 50: change to “these data were recorded”
Reviewer 2 Report
The manuscript of Marchese et al. deals with the very important issue of the efficacy of the semi-critical devices disinfection procedures.
The manuscript is well written and easy to read, though the data are few; may be to fill this gap authors duplicate tables and repeat results in figures, text and tables. No major observations can be done, but some suggestions
Abstract:
some typos
introduction:
line 72: please check “Istituto Mediterraneo para Trapianti and Terapie”
it’s quite unusual to anticipate the major result at the end of the introduction!
Results:
Tables 2 and 3 are redundant
In figure 3 the dots shouldn’t be connected by lines, since are independent measures (and the figure is uselessly complicated). However, the data represented in figure 3 are already detailed in tables 2 and 3, therefore the figure can be eliminated
Significances, as reported in the text, should be added in the tables
Conclusion
The conclusion session is a summary of the results. Please rewrite proper conclusions, include the limits and strength of the study, and the value and significance/perspectives of the study
Reviewer 3 Report
In the current manuscript, Marchese et al. reported an internal reprocessing protocol for controlling the contamination of Endoscopes. This study is very interesting and the data were presented properly.
I only have several concerns and would like to invite authors to perform a minor revision before the manuscript could be accepted.
1) Is there any other protocol about endoscopes sterilization currently adopted in other hospital, and what is the difference between your proposed protocol and those one(s)? (eg. efficiency? convenience?)
2) Indeed, the protocol proposed by authors already used an specific Endoscope washer-disinfector (Steelco EW2). What is the benefit of your new-developed protocol better than the standard protocol recommended by the Endoscope washer-disinfector?
